# Process Control Strategies in Chemical Looping Gasification—A Novel Process for the Production of Biofuels Allowing for Net Negative CO₂ Emissions

**Paul Dieringer *** , **Falko Marx, Falah Alobaid, Jochen Ströhle** and **Bernd Epple**

Institute for Energy Systems & Technology, Technical University Darmstadt, Otto-Berndt-Str. 2,
64287 Darmstadt, Germany; falko.marx@est.tu-darmstadt.de (F.M.); falah.alobaid@est.tu-darmstadt.de (F.A.);
jochen.stroehle@est.tu-darmstadt.de (J.S.); bernd.epple@est.tu-darmstadt.de (B.E.)
* Correspondence: paul.dieringer@est.tu-darmstadt.de; Tel.: +49-6151-16-22692

**Abstract:** Chemical looping gasification (CLG) is a novel gasification technique, allowing for the production of a nitrogen-free high calorific synthesis gas from solid hydrocarbon feedstocks, without requiring a costly air separation unit. Initial advances to better understand the CLG technology were made during first studies in lab and bench scale units and through basic process simulations. Yet, tailored process control strategies are required for larger CLG units, which are not equipped with auxiliary heating. Here, it becomes a demanding task to achieve autothermal CLG operation, for which stable reactor temperatures are obtained. This study presents two avenues to attain autothermal CLG behavior, established through equilibrium based process simulations. As a first approach, the dilution of active oxygen carrier materials with inert heat carriers to limit oxygen transport to the fuel reactor has been investigated. Secondly, the suitability of restricting the air flow to the air reactor in order to control the oxygen availability in the fuel reactor was examined. Process simulations show that both process control approaches facilitate controlled and de-coupled heat and oxygen transport between the two reactors of the chemical looping gasifier, thus allowing for efficient autothermal CLG operation. With the aim of inferring general guidelines on how CLG units have to be operated in order to achieve decent synthesis gas yields, different advantages and disadvantages associated to the two suggested process control strategies are discussed in detail and optimization avenues are presented.

**Keywords:** chemical looping; biomass gasification; process control; process simulation

## 1. Introduction

The reduction of greenhouse gas emissions (GHGE) in order to reach the unilateral goals agreed upon in the UNFCCC Paris Agreement is one of the major challenges of civilization in the 21st century. While notable advances in the energy sector have been achieved in recent years [1,2], the de-carbonization of the transport sector, which is responsible for almost one quarter of the European GHGE emissions [3] and consumes 36% of the global final energy [1], signifies a key issue on the path to a closed carbon cycle. Especially the replacement of conventional fuels in the heavy freight transport and aviation industry, where electrification is currently not viable, remains a major hurdle. When considering the European Union's Renewable Energy Directive (RED II) [4], which set a target of a share of 14% renewable energy in the transport sector by 2030, while at the same time alleviating negative impacts on food availability and prices, it is clear that significant advances in renewable fuel generation are required.

The production of so-called advanced or second-generation biofuels through thermochemical conversion of biomass-based residues is an auspicious pathway to achieve these goals. Gasification is

a mature thermochemical biomass conversion process, although its primary use is the generation of heat and electricity, while the synthesis of advanced biofuels through the gasification route has not been implemented in an industrial scale, yet [5].

Commonly, biomass gasification is achieved through utilizing air or pure oxygen in the gasifier. Albeit, pure oxygen is typically used in gasification processes embedded in biomass-to-biofuel process chains, since a nitrogen-free, high calorific value syngas is required for fuel synthesis [6]. The provision of this oxygen requires an air separation unit (ASU), which is associated with high capital and operational costs, hence adversely affecting the energetic plant efficiency and process economics [6,7]. Alternatively, steam [8–10] or carbon dioxide [10–12] can be deployed as the gasification medium. Yet, either of the two suffers from slow gasification kinetics [6,13,14] and strong process endothermicity [6,15], limiting the process efficiency. To circumvent this, the dual fluidized bed gasification (DFBG) technology achieves feedstock gasification in two connected reactors; a gasifier in which steam gasification of the deployed feedstock is attained, and a combustor in which the residual char is combusted facilitating full char conversion and the provision of heat, which is transported to the gasifier using an inert circulating bed material [16–18].

A similar gasification concept allowing for decent fuel conversions, without requiring an ASU is the chemical looping gasification (CLG) process, where biomass gasification is also carried out in two separate reactors (see Figure 1) [15,19–22]. Just as the related chemical looping combustion (CLC) process, CLG is realized using two coupled fluidized bed reactors, in order to attain good heat and mass transport characteristics [21,23,24]. Here, steam or carbon dioxide provide bed fluidization and gasification (see Equations (1) and (2)) of the feedstock in the fuel reactor (FR) [15,24]. Additional oxygen for the partial (see Equation (3)) or full (see Equations (4)–(6)) oxidation of gaseous hydrocarbon species, enhancing gasification kinetics and reducing the process endothermicity, is supplied through a circulating oxygen carrier (OC, $Me_xO_y$) [19,21,24]. Furthermore, the homogeneous water gas shift (WGS) reaction (Equation (7)) takes place inside the gas phase.

$$C + CO_2 \rightarrow 2\,CO \tag{1}$$

$$C + H_2O \rightarrow CO + H_2 \tag{2}$$

$$Me_xO_y + CH_4 \rightarrow Me_xO_{y-1} + 2\,H_2 + CO \tag{3}$$

$$4Me_xO_y + CH_4 \rightarrow 4\,Me_xO_{y-1} + 2\,H_2O + CO_2 \tag{4}$$

$$Me_xO_y + CO \rightarrow Me_xO_{y-1} + CO_2 \tag{5}$$

$$Me_xO_y + H_2 \rightarrow Me_xO_{y-1} + H_2O \tag{6}$$

$$CO + H_2O \leftrightarrow H_2 + CO_2 \tag{7}$$

The required oxygen transport to the FR is facilitated through a repeated regeneration of the OC (see. Equation (8)) in the air reactor (AR) with oxygen contained in the inlet air [15,20,24]. Moreover, unconverted char is combusted in the air reactor (see. Equation (9)), leading to a full conversion of the deployed feedstock [23,25].

$$Me_xO_{y-1} + 0.5\,O_2 \rightarrow Me_xO_y \tag{8}$$

$$C + O_2 \rightarrow CO_2 \tag{9}$$

The latter reaction is generally undesired, as a high carbon conversion is targeted inside the FR, in order to maximize the carbon capture efficiency of the process [23,26,27]. In literature, carbon capture efficiencies in the range of 90–99% are reported for CLC [26,28,29]. As approximately one third of the carbon contained in the feedstock is transferred into the valorized end-product (e.g., liquid Fischer-Tropsch fuels) in process chains employing CLG for syngas generation, this means that up to 65% of the carbon contained in the feedstock can be captured and stored, constituting negative

emissions in case biogenic feedstocks are being employed. Yet, in reality figures falling short of this value can be expected, as a fraction of the feedstock carbon will be lost in the AR in the form of $CO_2$.

Apart from the oxygen transport, the continuous solid circulation between the two reactors provides the required heat transport from the AR, in which the exothermic re-oxidation of the OC occurs, to the FR, where the endothermic gasification reactions take place [15,19,23], thus allowing for stable elevated reactor temperatures.

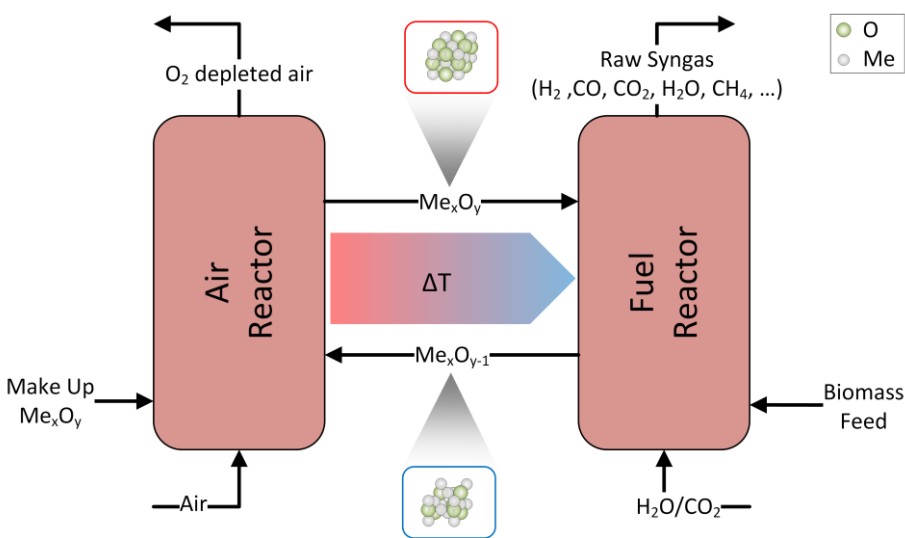

**Figure 1.** Schematic of chemical looping gasification (CLG) process.

CLG not only offers excellent characteristics in terms of feedstock flexibility [24], but is especially well suited for biomass-based feedstocks [30,31], commonly exhibiting a reactive char and containing a large fraction of volatiles. This means that high char conversions can be achieved through the gasification reaction with steam or $CO_2$, while volatiles are converted to the desired syngas species through their partial oxidation on the OC surface (see Equation (3)). Furthermore, it is reported that iron containing materials [32–35] can facilitate the cracking and oxidation of tars, which are known to be formed in significant amounts during biomass gasification [36].

While the role of the gasification agent is similar in CLC and CLG (i.e., char gasification), the oxygen carrier is meant to only partially oxidize the gaseous species in CLG, yielding a raw product gas with a high heating value [23,37], instead of a heat release from the AR, which is used for heat and power generation in CLC [24,38,39]. This shift from CLC to CLG is achieved through lowering the oxygen-to-fuel equivalence ratio in the FR to values below unity. An autothermal CLG process, maximizing the chemical energy contained in the raw syngas without relying on external heating, is obtained when the net heat release from the process equals zero (neglecting heat losses).

Although one might hence deduce that the transition from CLC to CLG is straightforward, there are major differences between the two processes. While large OC circulation rates are favorable in CLC, as they allow for a high oxygen availability in the FR, which favors fuel combustion [40–43] and provide for a large heat transport from the AR to the FR [41,44,45], the former is not desired in CLG. Here, the oxygen availability in the FR has to be limited in order to prevent the full oxidation of the employed feedstock. However, even more so than in CLC, CLG requires large heat transportation rates from the AR and FR due to the less pronounced occurrence of full oxidation reactions (Equations (4)–(6)), at the cost of highly endothermic partial oxidation reactions (Equation (3)) in the FR. This leads to a fundamental challenge in terms of process control, as both, heat and oxygen transfer between the two reactors, have to be controlled independently in order to attain an autothermal CLG process. Initial advances to reach this target were made by Ge et al. [37], diluting an active OC material with an inert, thus obtaining stable reactor temperatures for a lab-scale CLG unit. Yet, due to the significance of this inherent challenge, an in-depth analysis of this issue is required. Therefore, this work takes a holistic

approach to this matter, employing process simulations in order to establish suitable process control measures to attain an autothermal CLG process. In the following, the developed process model will be introduced in Section 2, before general process control and optimization strategies are presented and discussed in detail in Section 3. To round off these elucidations, the most crucial findings and an outlook on future research topics are given in Section 4 of this article.

## 2. Modelling Methods

### 2.1. Description of the Process Model

The deployed Aspen Plus™ model, shown in Figure 2, is largely adopted from a previous study by Ohlemüller et al. [25]. Here, the chemical reactions occurring in the AR and FR are modelled in two separate reactors, whereas gas-solid and solid-solid separation is achieved through cyclones and separators, respectively. In order to reduce model complexity, the AR and FR were modelled as equilibrium RGIBBS reactors in this work, as this simplification allows for a basic description of the most crucial phenomena required for process control and obviates the necessity of accurate kinetic data. To account for the solid circulation in chemical looping processes, a constant mass stream of solids continuously cycles through the system (OCR-TOAR/OCO-TOFR), after being added to the system after initiation of the simulation (INIT).

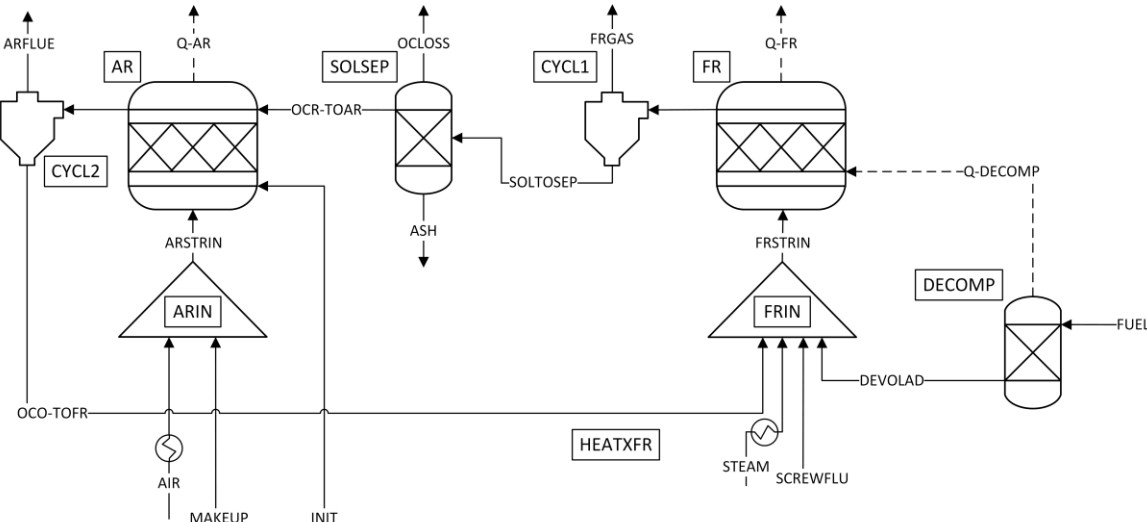

**Figure 2.** Flow sheet of the Aspen Plus™ CLG process model.

For completeness and comprehensibility reasons, all components and streams are briefly described in the following:

- Prior to any calculation, an initial solid mass flow is given into the system (INIT), to model the circulating solid OC mass. Instead of estimating the actual solid loss, the approach of Ohlemüller et al. [25], setting the total OC loss (OCLOSS) to 1% of the circulating mass to achieve fast flowsheet conversion, was adopted. The same amount of fresh solids was constantly fed to the AR (MAKEUP), to achieve constant solid circulation.
- For both reactors, cyclones are employed to achieve solid-gas separation. The FR products are separated into a gas (FRGAS) and solid (SOLTOSEP) stream, via CYCL1 (separation efficiency 100%). Similarly, the AR products are separated into a gas (ARFLUE) and solid stream (OCO-TOFR) in CYLC2 (separation efficiency 100%).
- All streams entering the process are fed at ambient temperature ($T_0 = 25\,°C$), except for the stream STEAM, which is fed as saturated steam (120 °C).

- The steam and the air entering the FR/AR are preheated to a designated inlet temperature ($T_{air,AR}$, $T_{H2O,FR}$). If not stated otherwise, the inlet temperature of both streams (STEAM, AIR) when entering the FR/AR is set to 400 °C.
- As Aspen Plus™ is not equipped to handle solid fuels, the biomass feedstock (FUEL) is fed to the decomposer (DECOMP), where it is decomposed into its pyrolysis products (DEVOLAD). The heat of pyrolysis (Q-DECOMP) is transferred to the fuel reactor. A detailed description of the decomposer block is given in Section 2.2.
- The pyrolysis products (DEVOLAD), the gasification agent (STEAMX), the OC recycled from the AR (OCO-TOFR), and the $CO_2$ required for solid feeding and loop seal fluidization (SCREWFLU) are mixed (FRIN) before entering the fuel reactor.
- Subsequently, the educts entering the fuel reactor (FR) are converted into reaction products according to the chemical equilibrium at the given boundary conditions ($T_{FR}$, $P_{FR}$ = 1 atm).
- The solids leaving CYCL1 are separated into the OC fed to the AR (OCR-TOAR) and a stream containing carbon and ash (SOL) in the solids separation (SOLSEP). This separation signifies the removal of bed material (i.e., OC, ash and unconverted feedstock) from the FR via sluicing during operation. Additionally, a fraction of the oxygen carrier material is removed from the system (OCLOSS), to model OC losses via sluicing and attrition.
- The OC makeup stream (MAKEUP) and the inlet air (AIRX) are mixed (ARIN) before being fed to the AR.
- Inside the air reactor (AR) the reduced OC and the unreacted char react with the oxygen contained in the air according to the chemical equilibrium at the given boundary conditions ($T_{AR}$, $P_{AR}$ = 1 atm).

*2.2. Decomposer*

Generally, the conversion of a fuel during gasification is described by three subsequent mechanisms: drying, pyrolysis and gasification [6]. While the gasification step is modelled in the FR, the former two mechanisms are modelled in the decomposer block in this study. As drying solely encompasses the release of moisture from the fuel [6,46], the main focus of this section is placed on fuel pyrolysis. Ohlemüller et al. [25] applied the pyrolysis model of Matthesius et al. [47] to predict the pyrolysis product composition from coal proximate and ultimate analysis parameters. Although it is reported that the basic mechanism of coal and biomass pyrolysis are similar [6,7], it was decided to employ a pyrolysis model specifically tailored for biomass feedstocks, as this study is focused on the conversion of biomass-based fuels. Neves et al. [48] devised a pyrolysis model for biomass feedstock built on the basis of an extensive experimental database. Similar to the pyrolysis model by Matthesius et al. [47], this model solely requires information on the feedstock composition (C, H, O and char content) to estimate the final chemical composition of the organics after pyrolysis, allowing for its straight forward implementation into the existing Aspen Plus™ model. Cuadrat et al. [49] found that the formation of tar and larger hydrocarbons (>C1) is negligible in the presence of ilmenite and steam/$CO_2$. Therefore, the assumption by Ohlemüller et al. [25] and Mendiara et al. [50] that tars and larger hydrocarbons are directly converted to methane and carbon monoxide was also adopted in this study. Moreover, oxygen and hydrogen contained in the char were converted to syngas, resulting in a char solely consisting of carbon. As the FR is modelled based on chemical equilibrium, these simplifications do not have an impact on the final simulation results.

By applying these assumptions, the product compositions after pyrolysis were calculated on the basis of the proximate and ultimate analysis of wood pellets, being the model feedstock for all subsequent considerations (see Table 1).

**Table 1.** Summary of the Ultimate and Proximate analysis for industrial wood pellets.

| Ultimate Analysis | wt-% | Proximate Analysis | wt-% |
|---|---|---|---|
| C (d.a.f.) | 50.8 | Moisture | 6.5 |
| H (d.a.f.) | 6 | Ash (d.b.) | 0.7 |
| N (d.a.f.) | 0.07 | Volatile matter (d.b.) | 85.1 |
| O (d.a.f.) | 43.2 | Fixed carbon (d.b.) | 14.2 |
| S (d.a.f.) | 0.008 | | |
| Cl (d.a.f.) | 0.006 | | |
| Net calorific value [MJ/kg] | 17.96 | | |

Since the pyrolysis product composition is highly temperature dependent [6,7,48], a constant temperature representing the FR temperature during CLG was selected as the input for the pyrolysis model ($T_{devol.} = 900\ °C$). A summary of the final product composition after de-volatilization, which was implemented into the process model, is given in Table 2.

**Table 2.** Mass yields [wt-%] for DECOMP Aspen Plzus® block for industrial wood pellets according to pyrolysis model of Neves et al. [48] (T = 900 °C).

| Component | wt-% | Component | wt-% |
|---|---|---|---|
| ASH | 0.65 | $H_2O$ | 14.06 |
| CO | 55.20 | $N_2$ | 0.06 |
| C | 11.92 | $CO_2$ | 3.11 |
| $CH_4$ | 13.55 | $H_2S$ | 0.01 |
| $H_2$ | 1.43 | | |

## 2.3. Boundary Conditions

For all subsequent simulations, the biomass input was selected in such a way, that the thermal load, $P_{th}$, of the chemical looping gasifier amounted to 1 MW. In terms of the circulating solid materials, the deployed oxygen carrier material is ilmenite, for which is has been established that the major redox stages are $FeO + TiO_2$, $Fe_3O_4$, $TiO_2$ and $Fe_2TiO_5$ [51]. These redox stages were modelled as $FeTiO_3$ (for $FeO + TiO_2$), $Fe_3O_4$, $TiO_2$, and $Fe_2O_3 + TiO_2$ (for $Fe_2TiO_5$). Deeper redox stages (e.g., FeO) were also considered in the process model, yet were not found to be formed in notable amounts. The inert solid sand was modelled through pure $SiO_2$. The FR and AR are operated under atmospheric pressure. Moreover, the air reactor temperature was set to 1050 °C, if not stated otherwise. The fuel reactor temperature results from the energy balance of the process, requiring that both reactors are in heat balance ($\dot{Q}_{FR} = 0$, $\dot{Q}_{AR} \geq 0$). As the kinetic syngas inhibition of char gasification reactions [8,12] is not considered in the RGIBBS equilibrium calculation, full char conversion is attained inside the FR for all temperatures considered in this study. Although this simplification signifies a deviation from reality, it does not impact the general inferences which will be elaborated on hereinafter. For the steam to biomass ratio in the FR a value of 0.9, reported for a 2–4 MW$_{th}$ chemical looping gasifier in literature [52], was selected if not stated otherwise. During CLC/CLG operation $CO_2$ is required for fuel feeding and inerting. This stream of $CO_2$, entering the fuel reactor, was selected in such a way that the $CO_2$ to biomass ratio amounts to 0.2, to take into account that the $CO_2$ input through the feeding section increases with increased thermal load. The two remaining process variables, the air mass flow entering the AR and the circulating oxygen carrier mass, were adjusted in such a way that autothermal CLG operation was achieved. A summary of all boundary conditions is given in Table A1 in Appendix A.

## 3. Results and Discussion

### 3.1. Attaining CLG Behavior

Generally, shifting from a combustion to a gasification process is achieved through lowering the air/oxygen-to-fuel ratio of the process, thereby decreasing the ratio of fully to partially oxidized gas species leaving the process and hence increasing the heating value of the product gas [6,53,54]. Here, the critical parameter is the so called air-to-fuel equivalence ratio given by the ratio of oxygen fed to the AR, $\dot{m}_{O,AR}$, and the oxygen required for full feedstock combustion, $\dot{m}_{O,stoich}$:

$$\lambda = \frac{\dot{m}_{O,AR}}{\dot{m}_{O,stoich}}. \tag{10}$$

According to this definition, (close to) full combustion of the feedstock is attained for air-to-fuel equivalence ratios larger than unity ($\lambda > 1$), while gasification processes require sub-stoichiometric oxygen feeding (i.e., $\lambda < 1$).

Due to the dissection of the gasification/combustion reaction into two separate reactors in chemical looping processes, there is no direct contact between the air entering the AR and the fuel entering the FR. Hence, the application of an alternative parameter, the oxygen-carrier-to-fuel equivalence ratio, $\phi'$, relating the amount of oxygen carried by the OC to the FR to the oxygen required for stoichiometric combustion, has been suggested [43]:

$$\phi' = \frac{R_{OC}\cdot\dot{m}_{OC}}{\dot{m}_{O,stoich}}. \tag{11}$$

Here, $R_{OC}$ denotes the oxygen transport capacity of the given oxygen carrier material. While this parameter accurately relates the two quantities for CLC, where the OC always leaves the AR in a (close to) fully oxidized state, this is not necessarily the case in CLG. Therefore, a slightly altered oxygen-carrier-to-fuel equivalence ratio, $\phi$, considering the possibility of a partially reduced OC leaving the AR, has been proposed for gasification applications [35]:

$$\phi = \frac{R_{OC}\cdot\dot{m}_{OC}\cdot X_{s,AR}}{\dot{m}_{O,stoich}}, \tag{12}$$

where $X_{s,AR}$ signifies the oxidation degree of the oxygen carrier at the AR outlet, given by [24,35]:

$$X_{s,AR} = \frac{m_{OC,AR} - m_{OC,red}}{R_{OC}\cdot m_{OC,ox}}. \tag{13}$$

Here, $m_{OC,red}$ and $m_{OC,ox}$ are the mass of an OC sample in a fully reduced and oxidized state respectively, while $m_{OC,AR}$ is the mass of the OC sample leaving the AR. For ilmenite the fully reduced oxygen carrier is approximated by $FeTiO_3$, the fully oxidized state is approximated by $Fe_2O_3 + 2TiO_2$, and $Fe_3O_4 + 3TiO_2$ denotes an intermediate redox state ($X_s = 0.67$).

In order to assess how $\lambda$ and $\phi$ have to be adjusted in order to obtain an efficient CLG process, one should first assess the general impact of these two parameters on the process. Due to the relative fast kinetics of the OC re-oxidation [55–57], the oxygen carrier is often assumed to leave the AR in a (close to) fully oxidized state for $\lambda > 1$ in chemical looping processes. In contrast, sub-stoichiometric air-to-fuel equivalence ratios ($\lambda < 1$) only lead to a partial re-oxidation of the OC in the AR. Following the same logic, the OC material can be assumed to leave the FR in a (close to) fully reduced state in case $\phi < 1$, whereas partial reduction is attained for $\phi > 1$. From these deductions, it becomes clear that "standard" CLC operation is attained for $\lambda > 1$ and $\phi > 1$, [42,43]. Here, a highly oxidized OC leaves the AR, before being partially reduced in the FR, which is illustrated in Figure 3a.

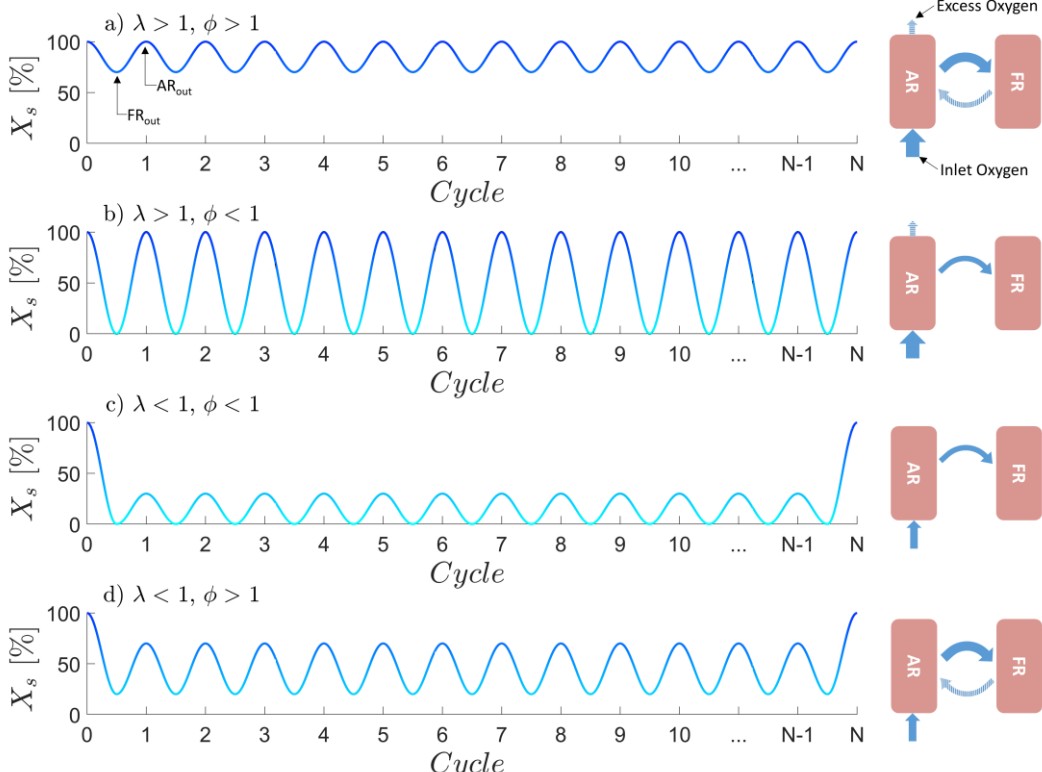

**Figure 3.** Different chemical looping modes (**a**–**d**) dependent on the air-to-fuel equivalence ratio $\lambda$ and the oxygen-carrier-to-fuel equivalence ratio $\phi$.

When targeting pronounced syngas formation, the oxygen release in the FR has to be limited, so that full feedstock oxidation is prevented [35,52]. The most obvious avenue that can be pursued to achieve this is lowering $\phi$ below unity. When doing so, the employed air-to-fuel equivalence ratio $\lambda$ determines how much oxygen is transported between the two reactors per gram of OC. In case of $\lambda > 1$, which is illustrated in Figure 3b, the oxygen carrier undergoes a full redox cycle and hence the full oxygen transport capacity of the OC material (i.e., $R_{OC}$) is exploited. On the other hand, $\lambda < 1$ means that in equilibrium the OC leaves the AR in a partially reduced state, hence also reducing the mass specific oxygen transport of the OC (see Figure 3c). Lastly, one might also consider a process with $\lambda < 1$ and $\phi > 1$, as shown Figure 3d. In order to attain a steady-state process exhibiting these characteristics, full reduction of the oxygen carrier has to be prevented in the FR (e.g., kinetically), so that a fraction of oxygen is transported back to the AR. This means that in contrast to the former approaches, this case cannot be attained in equilibrium-like conditions. While this approach might also be feasible for CLG operation in theory, straight forward measures allowing for a controlled oxygen release in the FR are not at hand. Consequently, lowering the oxygen-to-fuel-ratio in the FR (i.e., $\phi < 1$) is the most promising avenue to attain CLG behavior. When aiming for large syngas yields, $\phi$ has to assume values below unity, while values exceeding unity are targeted in CLC [42,43]. In the following, different effective control strategies to achieve this reduction in $\phi$, required for pronounced syngas formation in the FR, while at the same time achieving an autothermal process, will be investigated.

In order to simplify the subsequent considerations, a standard parameter to describe gasification processes, the cold gas efficiency (CGE), $\eta_{CG}$, will be deployed hereinafter. It describes which amount of chemical energy from the fuel is transferred to the gaseous FR product gas during gasification [6,7].

$$\eta_{CG} = \frac{\dot{n}_{gas,FR} \cdot (x_{CH4,FR} \cdot LHV_{CH4} + x_{CO,FR} \cdot LHV_{CO,FR} + x_{H2,FR} \cdot LHV_{H2})}{\dot{m}_{fuel} \cdot LHV_{fuel}} \qquad (14)$$

Here, $\dot{n}_{gas,FR}$ and $\dot{m}_{fuel}$ denote the mole flow of the product gas stream and the fuel input into the FR, respectively. *LHV* is the lower heating value of the fuel (mass basis) and the gas species (molar basis) and $x_i$ is the mole fraction of the gas species.

### 3.2. Reduction of OC Circulation

One approach to obtain CLG behavior, which has been suggested by Pissot et al. [52], is reducing the amount of OC cycled through the system ($\dot{m}_{OC}$), hence reducing $\phi$. This approach can be deduced directly from Equation (12). Due to the resulting lower oxygen transport to the FR, syngas formation is favored, as less oxygen for full oxidation of the feedstock is provided by the OC. The simulation results for this approach are given in Figure 4. When considering the gas composition (Figure 4a) of the streams leaving the air and fuel reactor, various trends are visible. As expected, the syngas content in the gaseous FR products increases with decreasing OC circulation rate, which can directly be attributed to the lower oxygen/fuel ratio in the FR. Consequently, steam and $CO_2$ formation decrease. Yet, it has to be noted that substantial syngas concentrations are only attained for $\phi < 1$, which requires significant reductions in the OC circulation rate, when compared to CLC, where OC-to-fuel equivalence ratios as high as 8 [27] and 25 [40] are reported in literature for solid and gaseous fuels, respectively. For the gas concentrations leaving the AR, a strong impact of $\phi$ on the effluent oxygen is visible. As the inlet air mass flow was not varied ($\lambda = 1.2$), this observation is clear, as less $O_2$ is removed from the gas stream due to the lower OC circulation for $\phi < 1$. Furthermore, the $CO_2$ content in the AR product is predicted to be insignificant, indicating a complete char conversion, which is expected in chemical equilibrium. When considering Figure 4b, showing the solid composition after the fuel and air reactor, it can be seen that the OC leaves the AR and FR in a fully oxidized ($Fe_2O_3$ + $TiO_2$) and reduced ($FeTiO_3$) state, respectively for $\phi < 1$, whereas the OC is only partially reduced (indicated through the presence of $Fe_3O_4$) in the FR in case $\phi$ exceeds unity. Hence, the fraction of $FeTiO_3$ leaving the FR strongly increases with decreasing OC circulation, signifying a higher degree of reduction of the OC, due to the lower oxygen availability. As expected one consequently obtains chemical looping combustion behavior (see Figure 3a) for oxygen-carrier-to-fuel equivalence ratios greater than unity ($\phi > 1$), whereas chemical looping gasification behavior (see Figure 3b) is attained for $\phi < 1$.

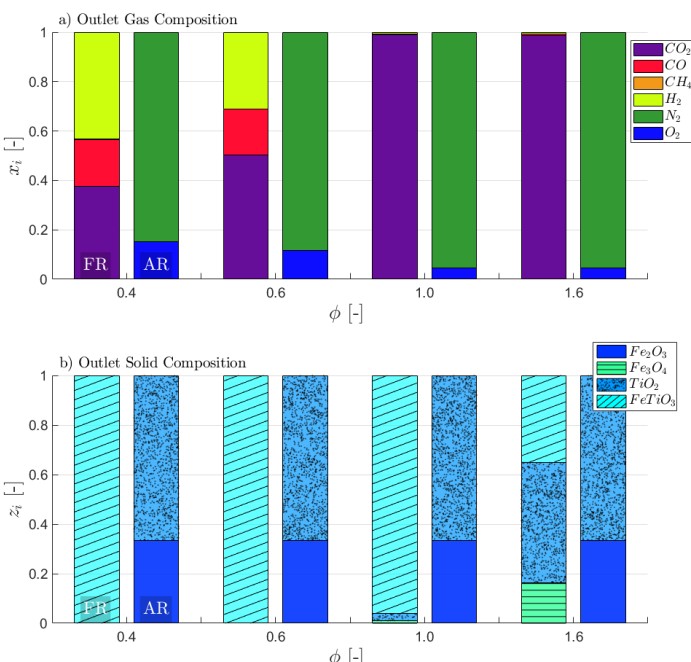

**Figure 4.** Simulation results for CLG operation through reduced oxygen carrier (OC) circulation. Dry molar gas composition (**a**) and molar solid composition (**b**) as a function of $\phi$ for varying OC circulation rates ($\lambda = 1.2$).

Based on these findings, one can conclude that a successful shifting from CLC to CLG for a given air-to-fuel ratio can be attained through a reduction in the OC circulation, which can also be seen in Figure 5a, showing a linear dependence between the two parameters. This means that for a change of $\phi$ from 1.0 to 0.5, the OC circulation rate has to be halved. However, lower solid circulation rates also result in a proportional decrease in the heat transport from the AR to the FR and hence a drop-off in FR temperatures [35,58]. While a moderate decrease in fuel reactor temperatures with decreasing OC circulation rate is visible for $\phi > 1$, for which complete feedstock conversion is attained in the FR, this decrease becomes more prominent for $\phi < 1$, where gasification reactions in the FR are dominant, hence increasing the endothermicity of reactions occurring in the FR. Consequently, FR temperatures fall below 800 °C for $\phi < 0.5$, where the availability of circulating OC material for sensible heat transport between the FR and AR is halved, when compared to $\phi = 1$ and more importantly the syngas content in the FR products is significant (see Figure 4a). This increase in syngas content also goes in hand with a decrease in the total net heat release from the CLG process ($\dot{Q}_{net}$), which can be calculated from the difference in the enthalpies of the streams entering ($_{in}$) and leaving ($_{out}$) the air and fuel reactor (see Equation (15)), as the enthalpy of the FR products increases.

$$\dot{Q}_{net} = \sum_{FR,in} \dot{m}_i \cdot h_i - \sum_{FR,out} \dot{m}_i \cdot h_i + \sum_{AR,in} \dot{m}_i \cdot h_i - \sum_{AR,out} \dot{m}_i \cdot h_i \qquad (15)$$

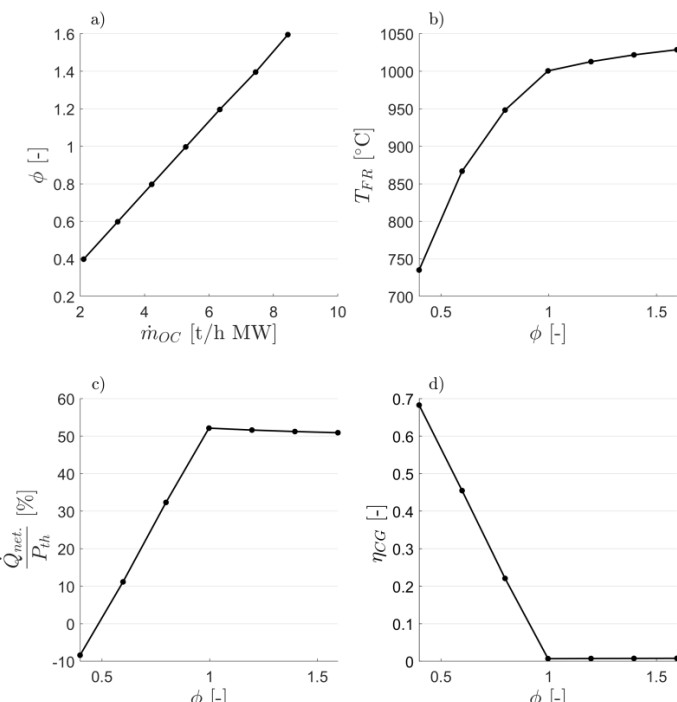

**Figure 5.** Simulation results for CLG operation through reduced OC circulation. OC-to-fuel ratio as a function of the OC circulation rate (**a**). Fuel reactor temperature (**b**), relative net process heat (**c**), and cold gas efficiency (**d**) for different values of $\phi$ ($\lambda = 1.2$).

The decrease in net process heat release with decreasing $\phi$, indicating the retaining of chemical energy in the FR products, also becomes visible upon consideration of Figure 5c, depicting the relative net heat release of the process for the different OC-to-fuel ratios. For the given boundary conditions, an autothermal process, for which syngas yields are maximized without relying on external heat addition ($\dot{Q}_{net} = 0$) is attained for an OC to fuel ratio of approx. 0.5. The resulting cold gas efficiency for this operating point amounts to approx. 60% (see Figure 5d) at a FR temperature of 775 °C. Although the equilibrium model predicts full char and volatile conversions for these temperatures (see Figures 4a

and 5d), char, volatile, and tar conversion are known to be kinetically governed processes in chemical looping systems [25,55,56,59], leading to product compositions deviating strongly from equilibrium composition [35,52]. Due to this reason, temperature differences in the range of 50 to 100 °C are generally targeted in dual fluidized bed gasification [16], in order to obtain sufficiently high gasifier temperatures, allowing for decent char, volatile, and tar conversions. Accordingly, FR temperatures in the range of 850–950 °C are desired in CLG, in order to attain high carbon capture efficiencies and cold gas efficiencies as well as low syngas tar loads [20,23,37,60,61].

These considerations underline that, although the desired reduction in $\phi$ is possible, attaining an efficient autothermal CLG process through a reduction in the OC circulation rate is not a recommendable strategy as it entails low fuel reactor temperatures, due to the dual-purpose of the OC circulation (i.e., oxygen and heat transport). Consequently, alternative approaches, allowing for a decoupling of oxygen and heat transport between the AR and FR and hence increased FR temperatures are required, in order to attain a CLG process exhibiting the desired characteristics.

### 3.3. Dilution of OC with Inert Bed Material

One strategy allowing for a decoupling of oxygen and heat transport between air and fuel reactor, which has been discussed in literature, is employing a mixture of an active OC material and a solid inert species (e.g., sand) [35,37,52]. Here, the inert fraction serves purely as a heat carrier, transferring sensible heat between the two reactors, without participating in the occurring reactions, while the active OC fraction fulfills its dual purpose of oxygen and heat transport. Consequently, this approach is a combination of CLG and dual fluidized bed gasification, which solely employs inert bed materials for heat transport. Following this logic, Ge et al. [37] found that through accurately tailoring the mixing ratio of inert silica sand and hematite, serving as an OC, FR temperatures can be stabilized at elevated levels (i.e., >900 °C), while at the same time ensuring a controlled oxygen transport to the FR, resulting in large syngas yields.

In terms of the impact of the variation in OC-to-fuel ratio on gas compositions achieved through this dilution of the OC material with an inert, similar observations are obtained (see Figure 6a). This means syngas formation increases steadily for $\phi < 1$. Moreover, the OC carrier composition, shown in Figure 3b, follows similar trends as observed for a plain reduction in the OC circulation rate (see Section 3.2), with a fully reduced OC leaving the FR for $\phi < 1$ (see Figure 3b), whereas only partial reduction is observed for $\phi > 1$ (see Figure 3a). Yet, the fraction of active OC material clearly decreases with decreasing $\phi$, due to the dilution with silica sand.

As the total amount of circulating solids is kept constant, the mass of circulating OC material is inversely proportional to the dilution factor. This means that there exists a linear relationship between the solid fraction of the inert material ($z_{SiO2}$) and $\phi$, which is visible in Figure 7a. Hence, for a given solid circulation rate, shifting from CLC to CLG can be attained through increased inert dilution. The positive effect of inert addition on FR temperatures becomes apparent upon consideration of Figure 7b. In contrast to a direct reduction in the OC circulation rate, the substitution of a fraction of the active metal oxide with an inert heat carrier allows for a sustaining of FR temperatures above 980 °C even for OC to fuel ratios as low as 0.5. Due to this increase in FR temperatures, the average temperature of the CLG process increases, leading to a slightly increased $\phi$ of approx. 0.55 for which autothermal operation is attained (see Figure 7c) (Higher process temperatures increase the heating demands of the educts entering the FR and AR and hence reduce the OC-to-fuel ratios for which autothermal operation can be obtained). Therefore, the cold gas efficiency obtained for autothermal operation for the given approach is also marginally reduced (see Figure 7d), when compared to the approach discussed in Section 3.2. Yet, it has to be noted that due to the intensified heat transport between the AR and FR, significantly smaller reactor temperature gradients are required for the given approach. Consequently, AR temperatures can be lowered without jeopardizing char conversions in the FR, thus reducing average process temperatures and allowing for strongly increased cold gas efficiencies (see also Section 3.5). Another advantage of this approach is that a catalytic material,

not participating in oxygen transport (e.g., olivine), could be employed for OC dilution instead of sand, allowing for improved syngas characteristics with regard to tar content.

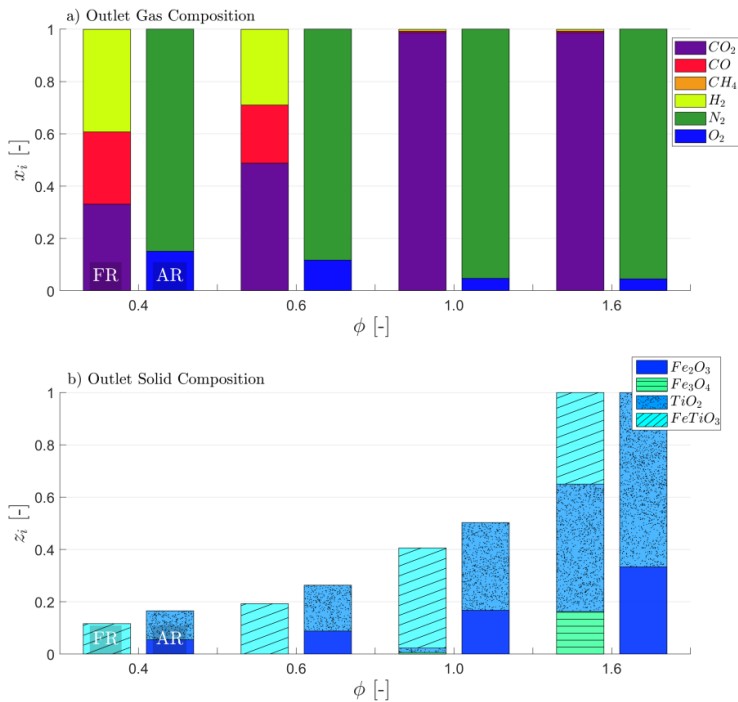

**Figure 6.** Simulation results for CLG operation through OC dilution with inert $SiO_2$ sand. Dry molar gas composition (**a**) and molar solid composition (**b**) as a function of $\phi$ for varying OC circulation rates ($\lambda = 1.2$, $\dot{m}_{OC} + \dot{m}_{SiO2} = const.$).

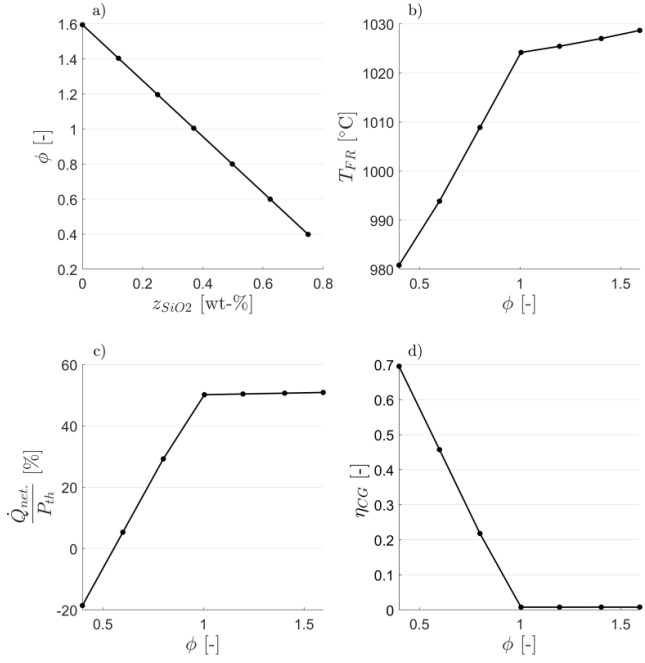

**Figure 7.** Simulation results for CLG operation through OC dilution with inert $SiO_2$ sand. OC-to-fuel ratio as a function of the inert concentration of the circulating solid mixture (**a**). Fuel reactor temperature (**b**), relative net process heat (**c**), and cold gas efficiency (**d**) for different values of $\phi$ ($\lambda = 1.2$, $\dot{m}_{OC} + \dot{m}_{SiO2} = const$).

Despite the presented advantages, Larsson et al. [35] found that, albeit slightly reducing tar loads, the addition of an active OC (ilmenite) to an inert circulating bed material in a dual-fluidized bed gasifier (for $\phi < 0.2$), entails a continuous drop in cold gas efficiency. This was explained by the fact that ilmenite addition does not enhance char conversion significantly, while its presence leads to a partial oxidation of the product gas. On the other hand, Pissot et al. [52] found that dilution of an active OC bed with up to 90% of an inert material does not entail visible enhancements in the cold gas efficiency of the CLG process, while it has a visible negative impact on carbon conversion. This shows that the mixing of an inert and an active OC material can have different effects on the process depending on the governing boundary conditions. Another drawback of this approach is that, albeit the addition of solids allows for an adjustment of $\phi$ during operation, it leads to a large system inertia, making it an arduous task to quickly react to disturbances. Moreover, a fraction of the solid material has to be removed from the system for ash removal in a continuously operated CLG unit. Economic considerations require a separation of these materials for further processing, recycling, and disposal. Clearly, the presence of a third component (i.e., sand, olivine) further complicates this task. Lastly, it is known that the operation of a fluidized bed with multiple bed materials of different characteristics brings about additional challenges in terms of material fluidization, entrainment, and attrition, as well as bed segregation [62]. Due to these reasons it was also suggested to employ materials of a low oxygen transport capability ($R_O$), such as LD-slag, containing a large inactive fraction not participating in the oxygen transport, which fulfills the purpose of the inert heat carrier [52]. Through this, oxygen carrier circulation rates providing sufficient heat transport between the reactors can be targeted, without obtaining OC-to-fuel equivalence ratios above unity. Yet, for this approach the main challenge is finding suitable OC materials exhibiting an oxygen transport capability in the desired range, high activity towards hydrocarbon conversion, and good chemical and mechanical stability.

### 3.4. Reduction of Air-to-Fuel Equivalence Ratio

To allow for a less restricted material selection and avoid solid inert addition, an alternative strategy to decouple oxygen and heat transport between the AR and FR is required. In order to achieve this, Larson et al. [35] suggested the deployment of a secondary system in which the OC is pre-reduced before entering the FR. This means that, as shown in Figure 3c, a partially reduced OC enters the FR ($X_s < 1$), thus entailing a lower OC-to-fuel ratio (see Equation (12)). Instead of employing a secondary reactor to accomplish this, one can also operate the AR in a sub-stoichiometric fashion ($\lambda < 1$), thereby preventing full re-oxidation of the OC in the AR. This means that in order to attain CLG conditions, the amount of air fed into the air reactor can be reduced, while retaining a constant OC circulation. As a consequence, the OC steadily reaches a lower degree of oxidation, hence lowering its oxygen release in the FR, until steady state is reached (more details see Appendix B). This approach has already been pursued in a 140 kW$_{th}$ chemical looping reforming unit, employing methane as a fuel [44]. The suggested concept becomes more lucid when considering the simulation results shown in Figure 8. Clearly, the amount of fully reduced ilmenite leaving the air and fuel reactor increases when decreasing the air input into the AR for $\phi < 1$ (see Figure 8b). While the same is true for the solids leaving the FR for all presented CLG approaches, a strong increase in the FeTiO$_3$ and Fe$_3$O$_4$ content in the AR products is obtained when reducing $\lambda$ below unity. This can be explained by the fact that the oxygen available in the air reactor is insufficient to fully re-oxidize the OC, signified through an O$_2$–free product gas from the AR for $\phi < 1$ (see Figure 8a). Consequently, a pure stream of N$_2$ containing small concentrations of Argon and other minor compounds is produced in the AR [44]. Since substantial quantities of OC are cycled through the system in a fully reduced state, they effectively act as an inert, meaning that they transfer sensible heat, but do not participate in the occurring chemical reactions through oxygen release and uptake. However, in practice the reduced OC could potentially function as a catalytic site for tar cracking and methane reforming and favor the formation of syngas [32–35], thereby enhancing the process characteristics. Another advantage of the given approach is that an undiluted OC can be employed, which simplifies the required solid-gas and solid-solid (ash-OC-char) separation and the

operation of the CLG unit with regard to the fluidization behavior. Moreover, the net heat duty of the process can be tailored promptly and easily through an adjustment of the air flow to the AR, allowing for quick responses to disturbances (e.g., variations in feedstock composition).

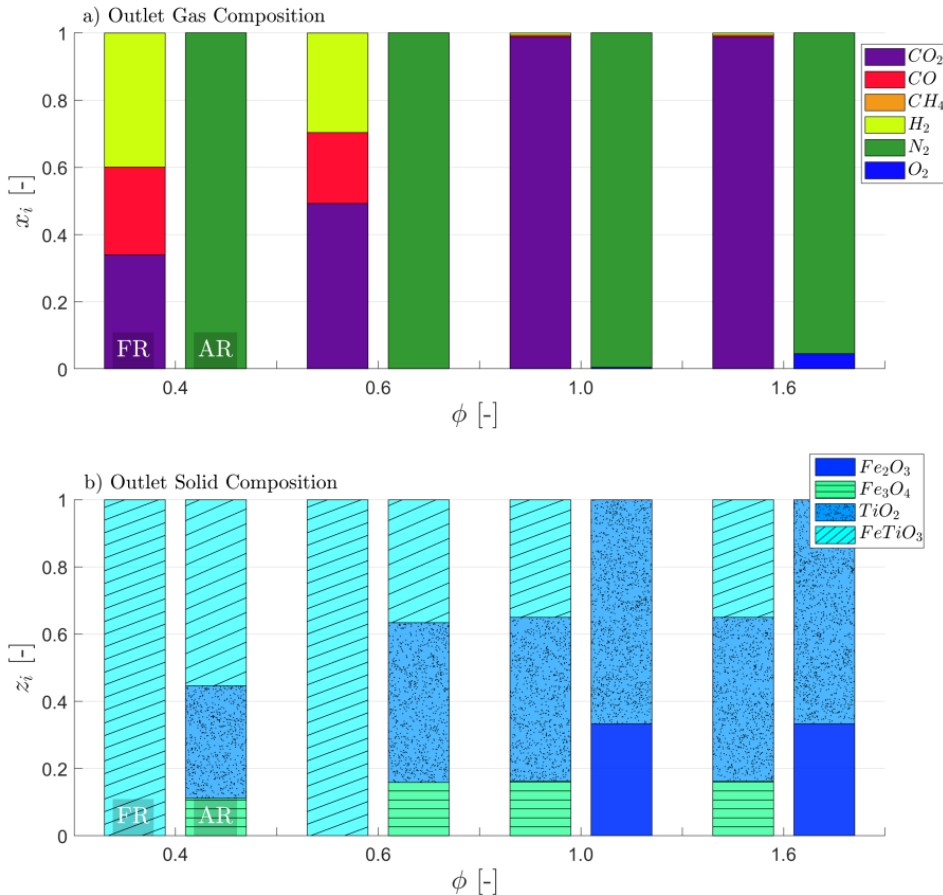

**Figure 8.** Simulation results for CLG operation through reducing $\lambda$. Dry molar gas composition (**a**) and molar solid composition (**b**) as a function of $\phi$ ($\dot{m}_{OC} = const.$).

The impact of the air-to-fuel equivalence ratio ($\lambda$) on $\phi$ is shown in Figure 9a. In CLC mode ($\lambda > 1$), where full OC oxidation is achieved in the AR (i.e., $X_{s,AR} = 1$), $\phi$ assumes a constant value, given by the amount of oxygen which is transported by a fully oxidized OC for a given circulation rate, regardless of the deployed air-to-fuel ratio (see Equation (12)). In contrast, lowering $\lambda$ to values below unity to attain CLG operation means that $\phi$ and $\lambda$ are equal, as the oxygen transport to the FR is limited by the oxygen availability in the AR:

$$\phi = \begin{cases} \lambda & \text{for } \lambda < 1 \\ \frac{R_{OC} \cdot \dot{m}_{OC}}{\dot{m}_{O,stoich}} = const. & \text{for } \lambda \geq 1 \end{cases} \tag{16}$$

The discontinuity of this relation for $\lambda = 1$ can be explained by the fact that when surpassing this value, a transient shift from CLC (see Figure 3a) to CLG (see Figure 3c) behavior (or vice versa) occurs, which goes in hand with a continuous decrease (resp. increase) in the oxidation degree of the oxygen carrier, before steady state sets in (more details see Appendix B).

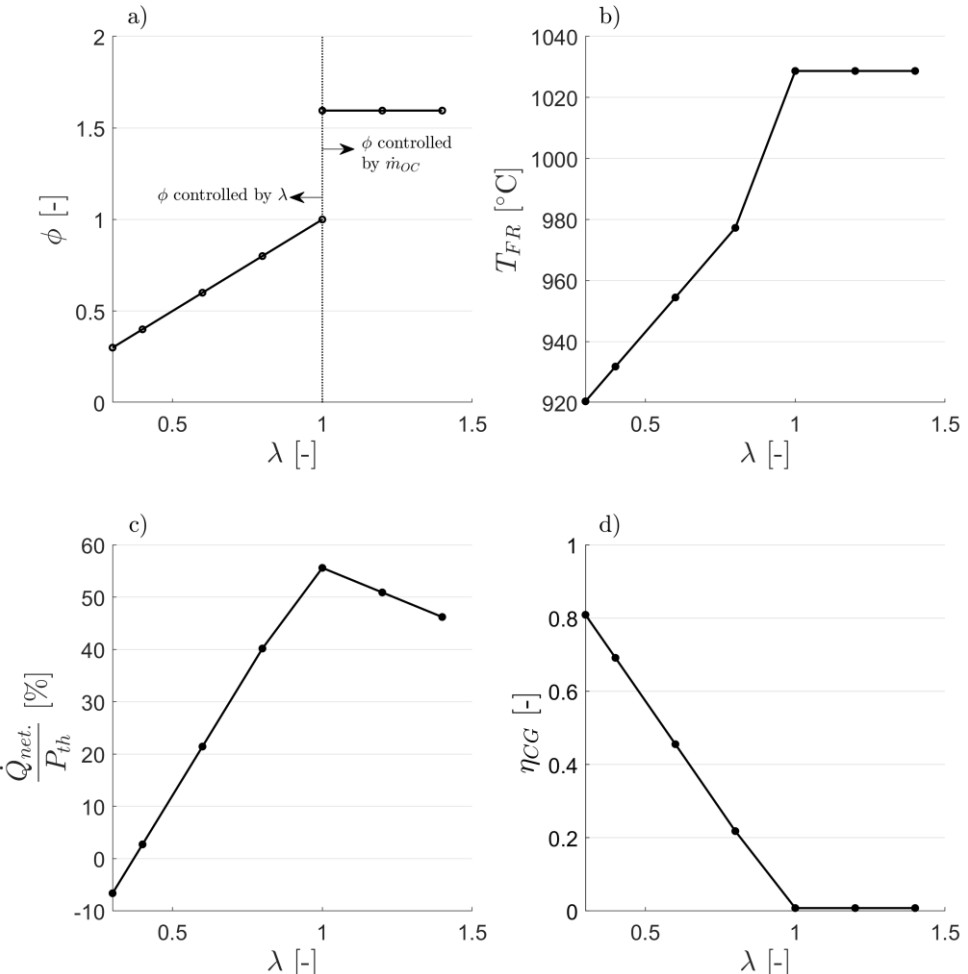

**Figure 9.** Simulation results for CLG operation through reducing $\lambda$. OC-to-fuel ratio as a function of the air-to-fuel equivalence ratio $\lambda$ (**a**). Fuel reactor temperature (**b**), relative net process heat (**c**), and cold gas efficiency (**d**) for different values of $\lambda$ ($\dot{m}_{OC} = const.$).

In terms of FR temperatures, Figure 9b shows that the given approach leads to a successful retaining of FR temperatures above 900 °C, even for $\phi$-values as low as 0.4, due to the transportation of sensible heat by the OC. Moreover, the given approach yields more beneficial results in terms of the process heat balance, which can be seen in Figure 9c. Clearly, autothermal CLG operation is attained for $\phi = 0.37$, which means cold gas efficiencies exceeding 70% can be achieved (see Figure 9d). This is the case as in contrast to the previous approaches (see Sections 3.2 and 3.3), the AR is not operated in air excess during CLG operation, reducing the loss of sensible heat through the AR off-gases. This means that if one would reduce the air feed to the AR to the minimum extent required for full OC re-oxidation for the CLG approach employing inert dilution (see Section 3.3), enhanced cold gas efficiencies could be attained. Nonetheless, the given approach clearly shows advantages in terms of process control due to its flexibility, the possibility of freely selecting a suitable OC material (i.e., no specific limits on $R_O$), without having to consider material mixtures, and the availability of a catalytically active reduced OC material, instead of an inert solid, cycling through the system. Moreover, the chemical strain on the OC material is reduced as the change in oxidation degree for each redox cycle is lower, when compared to the former approaches, relying on full reduction and oxidation in the FR and AR, respectively (see Figure 3b,c), which should have beneficial effects on the OC lifetime.

However, one issue that might arise due to the operation of the AR in an sub-stoichiometric fashion is related to the fact that during operation a fraction of the feedstock char leaves the FR unconverted and hence travels to the AR with the circulating OC material [23,26,27]. This so called "carbon slip"



leads to competing reactions between the OC material and the residual char, in case the AR is operated with $\lambda < 1$. Yet, simulations show that in an oxygen deficient atmosphere carbon conversion is favored to OC re-oxidation in chemical equilibrium. Moreover, CO formation shows to be negligible (more details see Appendix C). Due to the fast kinetics of both char conversion and OC re-oxidation, it can be expected that equilibrium-like conditions are attained in the AR and hence all residual char is fully oxidized to $CO_2$ in the AR. This hypothesis is also supported by chemical looping experiments in small scale fixed bed reactors, during which it was established that in the beginning of the re-oxidation stage oxygen preferentially reacts with deposited carbon before re-oxidizing the OC [21,63,64]. Nonetheless, experiments showing that this is also the case in a continuously operated CLG unit and that CO formation is negligible are required to establish that full char conversion without substantial CO formation in the AR can be attained for this approach. Another issue related to this approach is the potential deep reduction of the OC, which could potentially entail problems related to intensified OC attrition or bed agglomeration. Although the process model does not predict substantial formation of deeper reduction stages (e.g., FeO) in the FR, such phases, related to bed agglomeration, have been found to be formed in CLC under highly reducing conditions [51,65,66]. Therefore, the gravity of this issue should be further investigated in experimental studies.

## 3.5. Optimizing CLG Efficiency

In the previous section it was established that OC-to-fuel equivalence ratios smaller than unity are required in the FR. Moreover, it was demonstrated only when decoupling heat and oxygen transfer between the AR and FR, $\phi < 1$ and FR temperatures above 850 °C can be obtained for an autothermal CLG process. Thermodynamically speaking, it does not make a difference how this decoupling of heat and oxygen transport is attained, which is why the following considerations will focus on the CLG approach presented in Section 3.4, employing a reduction in the air-to-fuel equivalence ratio to achieve CLG behavior.

When optimizing gasification processes, the trade-off between maximizing the carbon conversion in the gasifier and at the same time attaining high cold gas efficiencies is at the core of many optimization strategies. This is also the case in CLG, where $\eta_{CGE} = 1$ and complete char conversion is desired, yet not attainable. While large carbon capture efficiencies are obtained in cases where the char is gasified in the fuel reactor to a large extent, which is promoted by high FR temperatures [20,23,37], large steam/biomass ratios [20,37], and high OC-to-fuel ratios (if sufficient char residence times are provided) [27,52], cold gas efficiencies are maximized by the minimization of the oxidation of $H_2$ and hydrocarbons in the FR [35]. Although full oxidation of syngas in the FR should be limited to achieve large CGEs, formation of steam and $CO_2$ in the FR is required to a certain extent to obtain autothermal CLG conditions. The degree to which this formation of fully oxidized gas species is required is determined by the criterion of the CLG process being in heat balance ($\dot{Q}_{net} = 0$). This means that the heat release attained through full feedstock oxidation has to balance the heat demand of pre-heating of all inlet streams to the given reactor temperatures, the heat of reaction for endothermic gasification reactions, and the heat losses of the CLG unit. This has also been shown in the previous sections where despite assuming chemical equilibrium (i.e., full feedstock conversion), cold gas efficiencies deviating strongly from unity were obtained for autothermal boundary conditions (see Figures 5, 7 and 9).

Therefore, one approach to enhance the cold gas efficiency in CLG is a reduction in the inlet gas flows entering the air and fuel reactor. Since the air mass flow entering the AR is required to control $\phi$, this leaves the steam mass flow entering the FR as a free variable which can be altered to enhance cold gas efficiencies. The effect of a reduction in the steam to biomass ratio on the net heat release of the process is shown in Figure 10a. It is visible that, with a decreasing steam to biomass ratio, the air-to-fuel equivalence ratio for which an autothermal process is attained decreases. Due to the direct correlation between the oxygen availability and cold gas efficiency in CLG (see Figure 10b), this also means that the CGE obtained for autothermal operation increases with decreasing steam/biomass ratio, so that the CGE is raised from 72.5 to 77.1%, when decreasing the steam/biomass ratio from 0.9 to 0.3. However,

it is obvious that the reduction of the steam to biomass ratio would also entail a drop in carbon capture efficiencies of the process, as less steam is available for char gasification and the kinetic inhibition effect of syngas increases with decreasing steam concentrations (entailing larger syngas partial pressures) in the FR [8,12,67,68]. This becomes most obvious for a steam to biomass ratio of 0, for which char conversions in the FR would be diminutive in a real gasifier, due to the slow kinetics of heterogeneous solid-solid OC-feedstock reactions [67–69]. As this drop in char conversion is not predicted by the equilibrium model, the negative effect on process efficiency with decreasing steam to biomass ratio cannot be evaluated in this study. However, sufficient steam availability clearly is a prerequisite in CLG, when targeting large char conversions and hence carbon capture efficiencies.

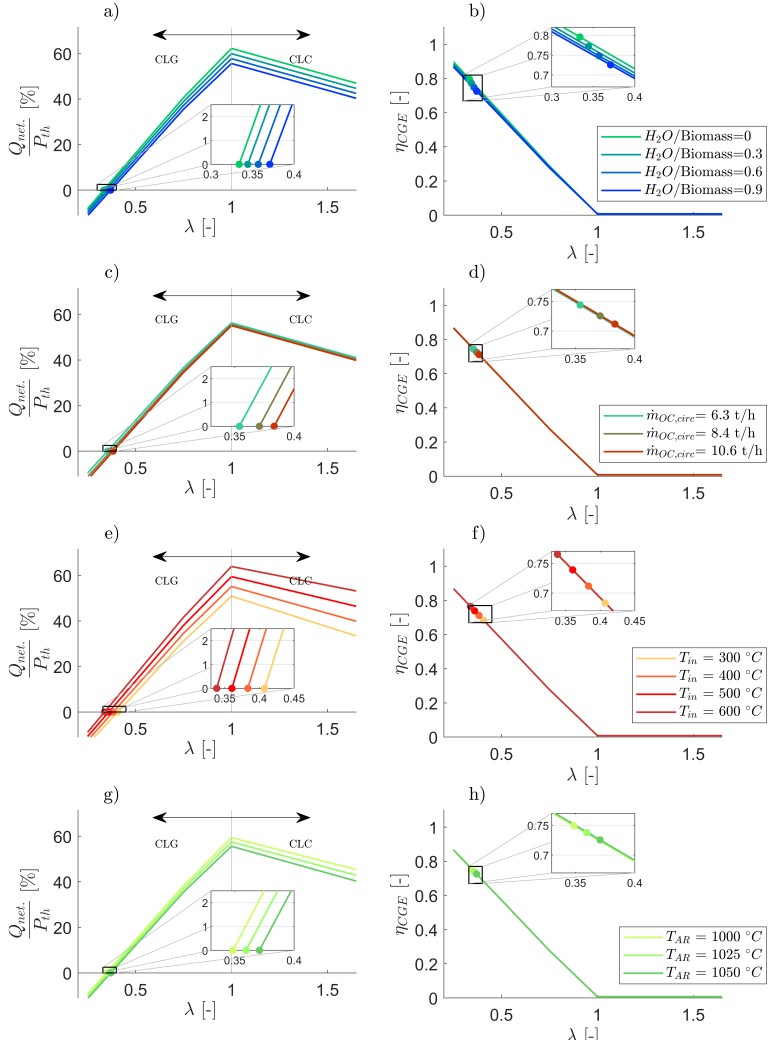

**Figure 10.** Net heat release and cold gas efficiency for CLC/CLG process as a function of the air to fuel equivalence ratio for different steam to biomass ratios (**a**,**b**), OC circulation rates (**c**,**d**), gas inlet temperatures (**e**,**f**), and air reactor temperatures (**g**,**h**). Circles mark the cold gas efficiency for autothermal CLG operation ($\dot{m}_{OC} = const.$, so $\phi = \lambda$ for $\lambda < 1$ and $\phi = const. > 1$ for $\lambda > 1$).

Another possible measure to enhance CGEs are variations in the circulation rate of the OC, which is shown in Figure 10c,d. Clearly, larger solid circulation rates enhance the heat transport between the reactors and hence entail higher FR temperatures [16]. However, due to material attrition, solid loss, which necessitates continuous make-up feeding, also scales with the circulation rate. As shown in Figure 10d, the effect of this material loss on the process heat balance is comparatively small, thus its effect on the cold gas efficiency is low. However, the model predicts an increase in FR temperatures

from 892 to 951 °C, when increasing the circulation rate from 6.3 to 10.6 t/h. This means that generally, large solid circulation rates are desired in CLG units, as large FR temperatures are beneficial for volatile and char conversion [20,23,37]. Yet, it has to be kept in mind that the solid circulation in dual fluidized bed systems requires solid entrainment from the fluidized bed riser, which can be increased through an increase in gas velocities (i.e., increase in steam/biomass ratio), smaller particle diameters or smaller reactor diameters [16]. Moreover, intensified solid circulation also increases the occurrence of a "carbon slip" to the AR, due to the lower residence times of the char particles in the FR [27,28,70]. This means that the OC circulation rate can only be varied within a given range.

Increasing the inlet temperature of the steam and air entering the FR and AR respectively, thereby decreasing the heat demand for heating up of the gases inside the reactor, is a further strategy to boost cold gas efficiencies. As shown in Figure 10e, this approach allows for a reduction of the air-to-fuel equivalence ratio from 0.38 to 0.34 when increasing inlet temperatures from 400 °C to 600 °C. Hence, maximizing inlet gas temperatures through heat recuperation is a key task in CLG in order to optimize the process efficiency, which is illustrated by the increase in the CGE from 68.3 to 76.5%, when increasing gas inlet temperatures from 300 to 600 °C (see Figure 10f). Due to the absence of corrosive compounds and the high process temperatures, the hot off-gases leaving the AR are ideal for steam generation and heat recuperation. On the other hand, special syngas coolers are being used to recuperate sensible heat from syngas streams for steam production [71–73], highlighting that efficient gas pre-heating using heat from process off-gases is possible in CLG.

Furthermore, variations in the AR temperatures can be considered, in order to enhance CLG process efficiencies. Generally speaking, a reduction in average process temperatures is beneficial for the process heat release, as pre-heating demands for all educts (i.e., inlet gases & feedstock material) are being reduced as a consequence, thus allowing for intensified heat extraction for a given air-to-fuel ratio (see Figure 10g). As visible in Figure 10h, a slight increase in the CGE by 2.4 percentage points can be attained for autothermal CLG operation when lowering AR temperatures from 1050 to 1000 °C. Yet, it has to be kept in mind that in chemical looping processes, air and fuel reactor temperatures are coupled, which means that a drop in FR temperatures is an inevitable effect of reduced AR temperatures. For the given boundary conditions, FR temperatures are projected to directly correlate with AR temperatures, which means that for the given reduction in AR temperatures from 1050 to 1000 °C, a corresponding drop in FR temperatures from 928 to 880 °C entails. This means that when attempting to prevent the ensuing drop in FR temperatures, related to negative effects on volatile and carbon conversion, OC circulation rates have to be increased accordingly as a counter-measure.

Although these insights allow for a first glimpse on process optimization approaches, it becomes clear that a detailed consideration of reaction kinetics and reactor hydrodynamics is quintessential, when aiming for a holistic optimization of the CLG process, as both phenomena have a pronounced effect on the process parameters. As it is well known that the conversion of char and other hydrocarbons is kinetically governed [25,55,56,59], the impact of reactor temperature, residence time, and gas concentrations on reaction kinetics need to be established in detail, allowing for accurate predictions of the governing reactions in a realistic environment. Moreover, reactor hydrodynamics are a crucial factor in chemical looping systems [74,75], making it a pre-requisite to consider them in advanced CLG process models. Through considering these phenomena, it thus becomes feasible to assess to which extent the preceding approaches can be utilized to obtain a CLG process exhibiting not only a high cold gas efficiency, but also excellent carbon capture efficiencies. Nonetheless, the preceding explanations offer valuable insights on the fundamental challenges associated with the autothermal CLG process, which require catering to, when implementing the technology in large scale.

## 4. Conclusions

In the course of this study, an equilibrium process model for the chemical looping gasification of biomass, using ilmenite ore as the oxygen carrier, was deployed to establish adequate process control techniques to attain autothermal behavior for gasifiers of any scale. It was shown that

pursuing continuous CLG operation leads to unique challenges in terms of the OC circulation, which is responsible for both, oxygen and heat transport between the air and fuel reactor. While high OC circulation is generally beneficial in CLC to achieve complete fuel conversion in the FR and prevent a drop in FR temperatures, CLG faces an essential dilemma. Here, large OC circulation rates are necessary to fulfill the process heat balance (i.e., retain constant temperatures in the FR), whereas significantly lower circulation rates are required in terms of the necessary oxygen transport. Hence, heat and oxygen transport have to be de-coupled. Based on model calculations, two strategies to achieve autothermal CLG behavior through a de-coupling of oxygen and heat transport were presented. One eligible option is the dilution of the OC with an inert solid (e.g., sand), allowing for an accurate tailoring of the mixture's heat capacity and oxygen transport capability through its composition. As an alternative, the oxygen transport to the FR can be controlled through the oxygen availability (i.e., air supply) in the AR, leading to a deeply reduced oxygen carrier cycling through the system, not being fully re-oxidized in the AR. While both approaches lead to stable autothermal CLG behavior with sufficiently high FR temperatures, the latter strategy possesses certain advantages in terms of process control and fuel reactor chemistry, based on which it was deemed more suitable for large-scale operation. Regardless of the deployed approach, it was shown that restricting oxygen release in the FR is key in controlling CLG operation, where large cold gas efficiencies are desired. As partial oxidation of the feedstock is necessary in order to fulfill the heat balance of an autothermal process, this means that heat losses and heat sinks in the chemical looping gasifier have to be minimized, so that the oxygen input into the FR can be reduced, thus boosting syngas yields. Possible strategies to achieve this are gas pre-heating, variations in the OC circulation, alterations in the average CLG process temperature, and a reduction in the $H_2O$/biomass ratio in the FR.

Certainly, the presented findings encourage a deeper investigation of the chemical looping gasification of biomass on a numerical level, as only through the deployment of elaborate models considering hydrodynamics and reaction kinetics in-depth inferences regarding the process efficiency are facilitated. Moreover, they also call for experimental investigations of the suggested process control strategies. Especially the suggested continuous CLG operation with a deeply reduced OC, not being fully re-oxidized in the AR, means setting foot on a new terrain. Here, the suitability of the presented approach is decided by the fact whether positive (e.g., pronounced methane reforming ability, increased syngas selectivity & tar cracking) or negative effects (e.g., intensified attrition, reactivity loss, particle agglomerations) prevail.

**Author Contributions:** Conceptualization, J.S. and P.D.; methodology, P.D.; writing—original draft preparation, P.D.; writing—review and editing, F.M., J.S. and F.A.; visualization, P.D.; supervision, B.E. All authors have read and agreed to the published version of the manuscript.

**Funding:** This work has received funding of the European Union's Horizon 2020-Research and Innovation Framework Programme under grant agreement No. 817841 (Chemical Looping gasification foR sustainAble production of biofuels-CLARA).

**Acknowledgments:** The authors would like to thank the Technical University of Darmstadt, enabling the open-access publication of this paper.

**Conflicts of Interest:** The authors declare no conflict of interest. The funders had no role in the design of the study; in the collection, analyses, or interpretation of data; in the writing of the manuscript, or in the decision to publish the results.

## Nomenclature

| Symbol | Explanation | Unit |
|---|---|---|
| $h_i$ | Enthalpy of stream $i$ | kJ/kg |
| $\dot{m}_i$ | Mass flow of component/element $i$ | kg/h |
| $M_i$ | Molar mass of component/element $i$ | g/mole |
| $\dot{n}_i$ | Mole flow of component/element $i$ | kmole/h |
| $P$ | Power | kW |
| $p$ | Pressure | bar |
| $R_{OC}$ | Oxygen transport capacity of oxygen carrier | - |
| $\dot{m}_{air,AR}$ | Mass flow of air entering the AR | kg/h |
| $T$ | Temperature | °C |
| $x_i$ | Mass/mole fraction in gas phase | - |
| $X_i$ | Conversion of component $i$ | - |
| $Y_{i,j}$ | Mass yield of component/element $i$ from substance $j$ | - |
| $z_i$ | Mass/mole fraction in solid phase | - |
| $\eta_{CC}$ | Carbon capture efficiency | - |
| $\eta_{CGE}$ | Cold gas efficiency | - |
| $\lambda$ | Air-to-fuel equivalence ratio | - |
| $\phi$ | Oxygen carrier-to-fuel equivalence ratio | - |

| Subscript | Explanation |
|---|---|
| AR | Air reactor |
| devol. | Devolatilization. |
| FR | Fuel reactor |
| init | Initial |
| net | net |
| O | Oxygen |
| OC | Oxygen Carrier |
| ox | Oxidation |
| red | Reduction |
| s | Solid |
| stoich | Stoichiometric |
| th | Thermal |

| Abbreviation | Explanation |
|---|---|
| AR | Air Reactor |
| ASU | Air Separation Unit |
| CGE | Cold Gas Efficiency |
| CLC | Chemical Looping Combustion |
| CLG | Chemical Looping Gasification |
| FR | Fuel Reactor |
| GHGE | Greenhouse Gas Emissions |
| LHV | Lower Heating Value |
| OC | Oxygen Carrier |
| RED II | European Union Renewable Energy Directive |
| WGS | Water-Gas-Shift |

## Appendix A. Boundary Conditions for CLG Process Model

A summary of all model boundary conditions employed for the simulations presented in Section 3.2, Section 3.3 and Section 3.4 is given in Table A1.

**Table A1.** Boundary conditions for 1 MW$_{th}$ CLC/CLG process model for different CLG approaches.

| Parameter | Approach 1 * | Approach 2 * | Approach 3 * | Unit |
|:---:|:---:|:---:|:---:|:---:|
| $T_{FR}$ | 730–1030 | 980–1030 | 930–1030 | °C |
| $T_{AR}$ | 1050 | 1050 | 1050 | °C |
| $p_{FR}/p_{AR}$ | 1.013 | 1.013 | 1.013 | bar |
| $\dot{m}_{fuel}$ | 200.4 | 200.4 | 200.4 | kg/h |
| $\dot{m}_{H2O,FR}$ | 180.4 | 180.4 | 180.4 | kg/h |
| $\dot{m}_{CO2,FR}$ | 40.1 | 40.1 | 40.1 | kg/h |
| $\dot{m}_{air,AR}$ | 1362.6 | 1362.6 | 454–1590 | kg/h |
| $T_{CO2,FR}$ | 25 | 25 | 25 | °C |
| $T_{H2O,FR}/T_{air,AR}$ | 400 | 400 | 400 | °C |
| $\dot{m}_{OC,init}$ | 2.11–8.45 | 8.45 | 8.45 | t/h |
| $z_{SiO2}$ | 0 | 0–75 | 0 | wt-% |

* CLG approach I: Reduction in OC circulation rate (see Section 3.2), CLG approach 2: Dilution with solid inert (see Section 3.3), CLG approach 3: Reduction of air inlet into AR (see Section 3.4).

## Appendix B. Shifting from CLC to CLG Operation through Variations in the Air-to-Fuel Equivalence Ratio

As described in Section 3.4, the oxygen availability in the FR is solely dependent on the circulation rate of the OC and the oxygen transport capability of the OC material ($R_O$), when operating the AR in air excess ($\lambda > 1$) in CLC, as the OC material is fully oxidized inside the AR. When subsequently reducing $\lambda$ to values below unity from a steady state CLC operating point (see Figure A1a), the limited air availability in the AR leads to a transient phase during which the OC undergoes a continuous drop in the oxidation degree with each redox cycle, as more oxygen is consumed in the FR (combustion conditions) than is being supplied in the AR. As soon as the oxidation degree in the FR approaches 0, the oxygen availability in the subsequent redox cycle is determined by the oxygen supply in the AR. Hence, $\phi$ is equal to $\lambda$ from this point onwards. As indicated in Figure A1a, this means that steady state CLG conditions are attained as a consequence. When on the other hand starting off with steady state CLG operation ($\lambda < 1$) before increasing $\lambda$ beyond unity, the OC undergoes a transient phase during which its oxidation degree increases with each redox cycle, since more oxygen is supplied in the AR than is being consumed in the FR. As soon as the amount of oxygen transported by the OC is sufficient to fully oxidize the deployed feedstock, CLC conditions are attained. It has to be noted that this can be the case before steady state is reached (see Figure A1b). This means that despite the described discontinuity in the relation between $\lambda$ and $\phi$ for $\lambda = 1$, a rapid switch in the OC-to-fuel ratio will not occur during operation, as the transition from CLC to CLG or vice versa will occur smoothly via a transient phase during which the oxidation degree of the OC adapts to the newly set boundary conditions.

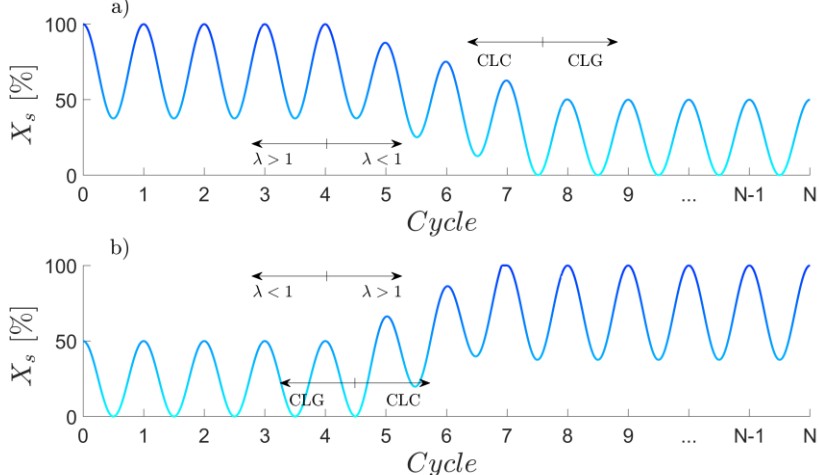

**Figure A1.** Progression of the OC oxidation degree when shifting from CLC ($\lambda > 1$) to CLG ($\lambda < 1$) mode through variations of the air-to-fuel equivalence ratio. (**a**) Shift from CLC to CLG, (**b**) shift from CLG to CLG.

### Appendix C. Char Conversion in an Sub-Stoichiometrically Operated AR

In order to establish how a mixture of unconverted char and a fully reduced OC behaves in an sub-stoichiometric oxygen containing atmosphere in the AR, a mixture of char (5 mole-%) and a reduced OC (78 mole-% $FeTiO_3$, 6 mole-% $Fe_2O_3$ and 11 mole-% $TiO_2$) were reacted with different amounts of air in an RGIBBS reactor of varying temperature (900–1100 °C). The results for an AR temperature of 1000 °C are shown in Figure A2. It is visible that char conversion occurs prior to OC re-oxidation, as the char fraction is zero regardless of the deployed air-to-fuel ratio. Moreover, the chemical equilibrium predicts a further reduction of the OC in case the amount of oxygen contained in the inlet air is insufficient for char conversion. Certainly, this behavior can only be observed in case of sufficiently long reaction times (rarely given in a fluidized bed), since solid-solid reactions between OC and char particles are known to exhibit slow kinetics [67–69]. This means that when attempting full char conversion, the inlet air entering the AR has to be sufficient to provide full carbon combustion. When this is the case, it can be assumed that full char conversion is attained inside the AR. In terms of the CO content at the reactor outlet it can be seen that full CO conversion to $CO_2$ is achieved regardless of the utilized air-to-fuel ratio, indicated by negligible concentrations of CO in the AR outlet (see Figure A2).

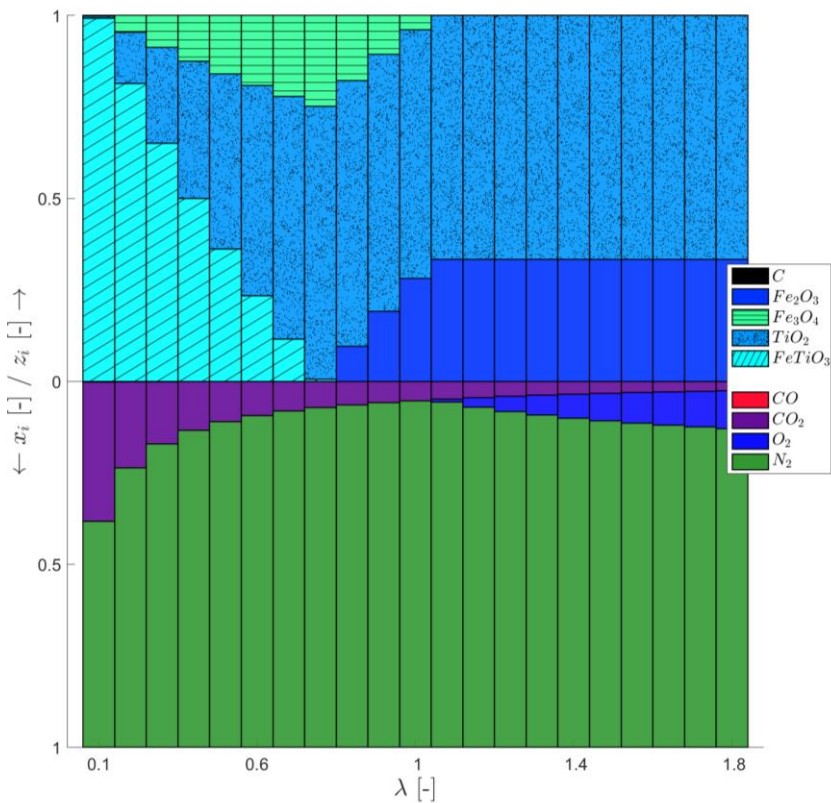

**Figure A2.** Solid and gas composition at chemical equilibrium for TAR = 1000 °C at varying air-to-fuel equivalence ratio $\lambda$ (Inlet solid composition: 78 mole-% $FeTiO_3$, 6 mole-% $Fe_2O_3$ and 11 mole-% $TiO_2$).

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
