# Peer review of "Process Control Strategies in Chemical Looping Gasification—A Novel Process for the Production of Biofuels Allowing for Net Negative CO2 Emissions"

_applsci, doi:10.3390/app10124271_

Round 1

Reviewer 1 Report

The whole simulation process starts from the wrong assumption that the biomass pyrolysis products that leave the decomposer (DECOMP) are only permanent gas and char.

It is well established and still under investigation, that during the pyrolysis of biomass, the gaseous products contain tars that depends of the size of biomass particles.

Primary catalysts, such as ilmenite, can reduce the amount of tars but its quantity it is not negligible, because there is pyrolysis gases that by-pass the solid particles with bubbles.

The thermodynamic equilibrium does not provide the presence of char at the temperatures considered in the paper, but experimentally it is always present.

In the paper missing data regarding the redox kinetics of ilmenite, and the production of fine as a function of the size of the particles.

As said by the authors in the conclusions, the theoretical simulation does not consider phenomena such as agglomeration due to the formation of low melting eutectics with consequent phenomenon of de-fluidization. Moreover, particle bed fragmentation and fines elutriation from the reactors have not be considered, neither the kinetics of char combustion in the Air Reactor, when partial oxidation of ilmenite is carried out, that it is a function of the char particle size.

The paper describe as a useful numerical exercise using the ASPEN PLUSTM model software.

Reviewer 2 Report

This study presents simulations results highlighting strategies to manage oxygen transport and heat transfer in chemical looping gasification. There are some limitations to the model, such as lack of reaction kinetics and hydrodynamics considerations, but these are clearly discussed by the authors. The simulations results offer interesting insights on reduction of the oxygen carrier circulation rate, diluting the oxygen carrier, and limiting oxidation of the oxygen carrier in the air reactor. I have included comments and edits in the attached document.  My main concerns are 1) The effect of CO2 and H2O entering the fuel reactor on oxygen demand from the oxygen carrier isn’t clear, and 2) there is no discussion on the possibility of FeO and Fe formation in the fuel reactor.

Round 2

Reviewer 1 Report

As specified by the authors, the article provides a very basic understanding of the CLG process, so as reported in my first comment, it can be considered as a useful simulation tutorial.

Reviewer 2 Report

The authors have addressed my concerns. Please note that in the pdf version I have access to, Figures 1, 2, 3 and 10 appear as empty boxes.